# The Accuracy of Commercially Available Fitness Trackers in Patients after Stroke

**DOI:** 10.3390/s22197392

**Published:** 2022-09-28

**Authors:** Anna Holubová, Eliška Malá, Kristýna Hoidekrová, Jakub Pětioký, Andrea Ďuriš, Jan Mužík

**Affiliations:** 1Faculty of Biomedical Engineering, Czech Technical University in Prague, 272 01 Kladno, Czech Republic; 2First Faculty of Medicine, Charles University and General University Hospital, 128 00 Prague, Czech Republic; 3Rehabilitation Centre Kladruby, 257 62 Kladruby, Czech Republic; 4Third Faculty of Medicine, Charles University in Prague, 100 00 Prague, Czech Republic

**Keywords:** activity tracker, physical activity, stroke, gait disorder, walking aid

## Abstract

**Background:** Fitness trackers could represent an easy-to-use and cheap tool for continuous tracking of physical activity of stroke survivors during the period of their recovery at home. The aim of the study was to examine the accuracy of the Fitbit activity tracker in locomotor activity monitoring of stroke survivors with respect to gait disorders, walking speed, walking aid, and placement of the tracker on body. **Methods:** Twenty-four ambulatory stroke survivors (15 men and 9 women) with locomotion/gait disorder were involved in the study. Patients underwent two walking tests with the Fitbit Alta HR trackers attached on 5 different places on body. The accuracy of the trackers has been analyzed on 3 groups of patients—those walking without any walking aid, those using a single-point stick and those using a rolling walker. **Results:** For no-aid patients, the most accurate place was the waist. Patients with a single-point stick revealed the smallest deviations for a tracker attached to a healthy lower limb, and patients with a rolling walker revealed the smallest deviations for a tracker attached on the paretic lower limb. **Conclusions:** An accuracy comparable with the healthy population can be reached for all of the three groups of patients, while fulfilling the conditions for minimum speed of 2 km/h and optimal placement of the trackers with respect to a walking aid and aspect to impairment.

## 1. Introduction

The use of fitness activity trackers (smart bracelets) in the healthcare sector is increasing due to their popularity, financial accessibility, wearing comfort and their motivational effect. Despite their use having already started in serious clinical patient monitoring, their accuracy is still uncertain. Implementation of these motivational tools for serious continuous monitoring of patients’ gait ability in their homes could increase the effectiveness of home-based rehabilitation, improve patients’ mobility and reduce periods of recovery [1,2,3,4].

Fitness trackers were designed and tested for the general adult healthy population, and their performance in people with impaired gait and using different walking aids is generally not known. Therefore, the accuracy of activity trackers may be insufficient for patients with different gaits [5].

Existing studies comparing the accuracy of activity trackers in patients after stroke [6,7,8] are limited to the use of special medical devices (e.g., ActiGraph or Metria-IH1) for clinical research purposes [9,10,11] or feature conditions of participants close to the healthy individuals [12].

Previous research suggested an error in step counting of the trackers up to 10% and less as being acceptable in free living conditions [13]. There are studies dealing with the accuracy of various fitness trackers in healthy people [14,15,16,17,18,19], some of them in conditions of performing low-intensity of physical activity [20], or in the elderly population with limited movement speed [21,22,23,24,25,26,27,28]. The average walking speed for the elderly people with a walking disorder is 1.22 km/h [29]. At speeds of around 1 km/h, errors of up to 95% occur [21]. Moreover, trackers at speeds below 0.5 m/s (1.8 km/h) could no longer detect more than a half of the actual steps being made [30]. Therefore, for this population the walking speed is a crucial factor determining the device’s (in)accuracy [31].

In other studies, where gait disorders were already present among participants, it was found that, in addition to speed, the device’s ability to detect steps was also affected by gait disorders, the use of a walking aid [32] and the location of the activity tracker on the body [33]. For example, walking with a rolling walker causes a specific body movement—mostly due to its handles being held by both upper limbs, which causes significant speed changes compared to the average human walking and also affects kinesiology of gait. In addition, if the activity tracker is attached to a paretic side, the step count accuracy is lower compared to the non-paretic side [30]. Patients move the rolling walker in front of them and the hands are placed in the same position. Another disturbing factor represents a terrain. Walking on stairs or on an inclined plane [29] can influence the accuracy as well.

Another factor is that conditions for steps to be recognized need to be fulfilled for some devices. For example, the ActivePAL (PAL Technologies Ltd., Glasgow, UK) activity tracker detects only the activity that takes more than 10 s continuously [34].

All of these limitations need to be considered before such devices are implemented into a real practice for home-based telemonitoring of patients after stroke.

Therefore, the aim of the presented study was to examine whether the commonly available fitness trackers, such as the model produced by Fitbit Inc. company, could be used for physical activity monitoring of patients after stroke. If so, the aim was to define the conditions for its possible use with respect to patient’s disorder, walking speed, walking aid, and body location.

## 2. Materials and Methods

### Experiment Methodology

Adult poststroke patients, who were hospitalized in the Kladruby Rehabilitation Centre during the period from August 2018 to June 2019 and met inclusion criteria, were included in this study (Figure 1).

Inclusion criteria contained acute and chronic patients after stroke, aged 18–70 years, with various gait disorders. Enrolled patients had to be able to walk a minimum of 100 m at their own speed, with or without walking aids.

Excluded were patients who were unable to understand the basic instructions (1). The measured trials in which the participants’ speed had been below 2 km/h were not included in the results (2).

Data of 24 patients (15 men) with left-sided (*n* = 14) or right-sided hemiparesis (*n* = 10) caused by stroke, aged on average 59 (SD 13) years, have been analyzed. Each participant signed the informed consent form before entering the study. The study has been approved by the Ethical committee of the Rehabilitation Center Kladruby (EK 20/2).

Each participant underwent two walking tests along a 70-m-long corridor. At the starting point, five activity trackers (Fitbit Alta HR) were used, each being placed on a participant’s upper left limb (ULL), upper right limb (URL), lower left limb (LLL), lower right limb (LRL) and the waist (WA) (Figure 1). Fitbit Alta HR detects steps based on change in speed. In the accelerometer inside the activity tracker, a change in speed causes a change in the resistance of the piezoelectric crystal, which the sensor then evaluates as either sufficient and adds a step or does not detect a step. In addition to steps, the Alta HR also measures heart rate, walked distance, energy used and active time. It can last for up to 7 days on a single charge.

During the walking test, the total walking time and the manually counted number of steps (using a tally counter) were measured. After the patient finished a test, the number of steps counted by the tracker was registered. Based on the patient’s gait disorder, a walking stick or a rolling walker was used during the test. Each patient was also supervised by a physiotherapist all along the route for security measures. If patients needed to stop, hold or sit, the trial was interrupted, and time and steps were not counted.

The data collected during the study included number of steps registered by the activity trackers, manually counted number of steps (using a tally counter), total time of walking, body location of the sensors, and walking aid used.

A comparison of the number of steps measured by the activity tracker at each place on the body with the actual number of steps measured by observing person using a tally counter the relative error has been performed. Friedman’s statistical test was used to prove dependence of accuracy on the body location. Due to the fact the speed below 1.8 km/h causes inaccuracy of more than 50% in fitness activity trackers [34], the data included in the analysis were only taken from the trails in which the participants walked with the speed of 2 km/h and more.

Since the acceptable error of the trackers is suggested to be up to 10% and less in free living conditions, the maximum acceptable deviation for our laboratory testing has been set up to 5%.

## 3. Results

A total of 48 trails (18 walks with a rolling walker, 14 walks with a single-point stick held in a healthy upper limb and 16 walks without any walking aid) were analyzed. Based on these data, relative deviations from the actual number of steps were calculated.

From the measured data, it was found that at speeds higher than 2.5 km/h, deviations greater than ± 5% occurred in 38% of cases throughout all the trackers on the body. Almost half of the inaccurate results (48% of the cases) were caused by a walking aid, a single-point stick and a rolling walker.

A Shapiro–Wilk normality test (α = 0.05) denied normality of the data, so a nonparametric Friedman’s test (α = 0.05) was used to assess the relation between body location and accuracy of the tracker.

Table 1 represents the results of the mean absolute relative difference (MARD) and standard deviation (SD) for each group according to a walking aid. (One group used no aid).

Graphical interpretation showing the most accurate position of the trackers for given placement on the body can be seen on Figure 2. Green spotlights indicate a position on which all the measurements fulfilled the condition for deviation lower than 5%. The yellow spotlights indicate position on which the average deviation throughout the measurements was lower than 5%, but some single measurements exceeded this level.

The Friedman’s test confirmed that the activity tracker accuracy depends on the body location (*p* = 0.000035 for the walk without any walking aids; *p* = 0.0009 for the walk with a single-point stick; *p* = 0.0001 for the walk with a rolling walker).

Figure 3 shows the relative deviations of all the trackers when walking without any aid, with a single-point stick and with a rolling walker.

Boxplots on Figure 4, Figure 5 and Figure 6 show relative deviations specifically for each of the three cases, i.e., (1) walking without any walking aid (Figure 4), (2) walking with a single-point stick (Figure 5) and (3) walking with a rolling walker (Figure 6).

## 4. Discussion

Results presented in this study show that the Fitbit Alta HR smart bracelet can be used for remote monitoring of gait ability in patients after stroke, but certain conditions need to be met. Due to the limitation of number of patients in specific groups differing in the walking aid used, conditions for patients who did not need any support for walking, those who used a stick and those who used a rolling walker were analyzed. Regarding the speed limitation, data of patients whose speed was at least 2 km/h were analyzed.

For patients who do not need to use any walking aid, the smart bracelets are the most accurate when attached at the waist (Figure 4), whereas the maximum measured relative error was 1.4%. Acceptable results were also found with a tracker attached to a healthy lower limb, where the maximum relative error for all the walks did not exceed 5%. Therefore, if the patient is able to walk without any aid, then the activity tracker should be worn and fastened at the waist or above the ankle of a healthy lower limb. However, it is necessary to create a strap for these two positions, as it is not supplied by the manufacturer. To confirm these preliminary results, a measurement of larger groups of patients needs to be made.

Regarding walking with a single-point stick, the smallest deviations were measured on a tracker attached to a healthy lower limb, where only one deviation exceeded 5%. On the contrary, the deviations on the healthy upper limb were higher compared to the paretic upper limb, which can be caused by the change in the movement of an arm on which the stick is being held. High deviations were also measured with a tracker attached to the waist (up to −69.68%). These deviations are probably caused by asymmetrical body movement when walking with a stick in one hand. The second most accurate position was measured on the paretic upper limb, with the MARD of 1.72% (SD = 7.40).

As a result, the relative error measured on the upper limbs was 100% in most cases (12 out of 18 for paretic upper limb and 15 out of 18 for healthy upper limb), meaning no step was detected by the sensor. On a healthy lower limb, the relative error exceeded 5% in four cases, while on a paretic lower limb only once. The high variance in both lower limbs was caused by an extreme value measured on both lower limbs of the same patient (151.33% in the healthy lower limb and −61.06% in the paretic limb). This abnormality could be caused by a specific gait of the patient; which was, however, not visibly observed during the measurement. Positive relative error means that the tracker counted more steps than the tally counter (reference value), which could be produced by shaking the limb or insufficient tightening of the tracker strap. The trackers attached to the paretic lower limb, healthy lower limb and the waist had comparable accuracy results.

While analyzing the impact of the step length or the cadence on the final accuracy, no significant dependence was observed. However, the accuracy is mainly limited by speed.

When considering the tracker to be used for home-based monitoring of patient’s daily physical activity, it is necessary to ensure the patient is able to walk the minimum speed of 2 km/h. It is also important to pay attention to the firm attachment of the tracker to maintain the required accuracy.

What also needs to be considered as a potential factor of inaccuracies is the limited number of participants in this study. To obtain more relevant data it would be necessary to measure at least 30 walks in each group. Another disruptive factor can represent a terrain. While in the presented study the participants were walking along a straight corridor without any obstacles, in free-living conditions different slope and surface of the ground can be presented. Therefore, determining accuracy in free-living conditions can be the subject of another study related to this research area. For that case, the error up to 10% could be accepted as sufficiently accurate for its use.

Several similar studies have been focused on accuracy of the commercially available activity trackers and although different in population or methodology, their results are similar. Mudge et al. used an ankle-placed StepWatch Activity Monitor [35]. They had a similar sample of 25 after-stroke patients but did not include any walking aid. Step counts from activity trackers were compared to the numbers from a foot switch. Patients walked various distances (8–250 m) indoors and outdoors. Similarly, accuracy of our findings on the non-paretic leg was 98.6%, and the error was 4.9% on the paretic one. Cesar et al. used Fitbit Charges on both wrists of 14 children and adolescents with impaired gait due to cerebral palsy (6), brain injury (5), stroke (2), and acute transverse myelitis (1). Six patients had bilateral involvement: two used unilateral walking aid. All patients were tested bya 30-step-test and a 6-min-test walk. They also found fair reliability of the Fitbit activity tracker (rs = 0.48; *p* = 0.01) and no significant difference between the two repeated attempts of the walking test [36]. Kooner et al. used nine healthy volunteers with Fitbit Zip (hip) and Fitbit Alta bracelet on the wrist and performed a 100 m walking test in a corridor. Probands used several walking aids (none, cane, walkers, etc.). Similar to us, the authors found that the error in many cases was less than 10% but, in some cases, especially while using walkers and wrist bands, can reach up to 90% [37]. Clay et al. used community-dwelling stroke survivors (average age 66 ± 8 years) and Fitbit Zip on the non-paretic hip side with and without a walking aid. Patients walked a six-minute walk test at a distance of 15 m [38]. They found very strong correlation between manually counted steps and steps counted by the Fitbit Zip (τ = 0.80; *p* < 0.001). They also found a very strong negative relation between the time spent walking and the accuracy. They set the minimum walking speed threshold to 0.8 m/s (2.88 km/h), which is very close to our recommendation of minimally 2 km/h

### Limits of Study

A limitation of the study was the placement of trackers on paretic upper limbs (spastic paresis, upper limb in an arm sling), where the measurement was not accurate. During walking there was no movement of the paretic upper limb, and no steps or too many steps were recorded (in the case of spasm of the spastic limb).

Another limitation could have been the device on the lower limb. The use of a device (splint, orthosis) on the lower limb was not included in the results. The most commonly used device was a peroneal tape, carbon or plastic splint. A device on the lower limb can affect both the stereotype and the gait speed. During testing it was not possible to remove the device from the lower limb because there would be an increased risk of falling, injury of an ankle or even the inability of the patient to walk.

Post-stroke patients with severe cognitive or phatic deficits were not included in the study due to the need to understand the instructions. All patients were consulted by a clinical speech therapist and a psychologist about suitability for inclusion.

This study was focused on the validation of a commercially existing device (Fitbit Alta), which was designed and tested on healthy individuals. Since the manufacturer (Fitbit Inc., now owned by Google LLC) has not published details about the algorithm for the step detection, we were not able to analyze the algorithm and discuss its characteristics. In general, the algorithm is based on hand-swing movement detection from a three-axis accelerometer, and allows not only step counting but also determines the frequency, duration, intensity, and patterns of your movement. There are also different approaches that could be used for step counting, for instance approaches based on the kinematic model presented in the study by Głowiński and Ptak [39].

## 5. Conclusions

In the presented study, both accuracy and best position of Fitbit Alta activity tracker on the body have been tested on patients after stroke with different gait mobility and different walking aids were used. If a patient is able to walk without any walking aid, the tracker is most accurate when being placed on the waist. At this position, the tracker achieves the accuracy of 100% for half of the measurements. The activity tracker placed above the ankle of a healthy lower limb also achieves high accuracy. At this position, the accuracy was 100% for 7 of 18 walks, and at the same time none of the deviations exceeded the limit of 5%.

In patients who use a single-point stick as a support when walking, the activity tracker is the most accurate when mounted above the ankle of a healthy lower limb. At this position, the deviation exceeded 5% in one of the 14 measurements. The data show that when fastened at the waist, the tracker is also sufficiently accurate at higher walking speeds. However, more measurements are required to confirm this finding.

When walking with a rolling walker, the activity tracker placed on the paretic lower limb proved to be the most accurate. The tracker at this point exceeded the 5% limit only once, in the case of a patient who had a high deviation even on the tracker on a healthy lower limb. The activity trackers on the healthy lower limb and waist were comparatively accurate. It may be necessary for people using the rolling walker to take this measurement before deciding on a suitable place to wear the tracker.

If the above conditions are observed, the measurement of steps is sufficiently accurate for use in rehabilitation care.

## Data Availability

The data presented in this study are available on request from the corresponding author. The data are not publicly available due to the privacy of probands.

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
