# Peer review of "The Accuracy of Commercially Available Fitness Trackers in Patients after Stroke"

_sensors, 2022, doi:10.3390/s22197392_

Round 1

Reviewer 1 Report

Dear Authors,

thank you very much for sending the article titled: The Accuracy of Commercially Available Fitness Trackers in Patients After Stroke for the review process. The approach of the study appears very original. The contents of the manuscript are quite interesting by his methodology and the tools used. Below are suggestions to the authors:

·        line 5 (What do you mean: "... the accuracy may be insufficient for patients with different walking gait [5]". Please describe it

·        please include the Table 1 with Patient description statistics

·        lack of system description (parameters)

·        typos errors

·        line 98. (What do you mean: "... If patient needed a physical help, the trial was interrupted and time and steps were not counted.") Please explain what help is meant

·        I suggest remove subtitles 2.1, 2.2, 2.3, 2.4, 2.5 and change it to 2.1 Experiment methodology

·        line 107. In what way do you calculate the error? Please describe it

·        given walking aid conditions??? what do you mean?

·        Table 1. For example, in Paretic UL variation coefficient (SD/MARD) (%) is very high 135% - No aid and 1,65%. High value denotes a large diversity of the trait and testifies to the heterogeneity of the studied population. There should not be such high values (the same in Rollator Paretic LL). Please describe it.

·        Figure 1. Please include a picture of the system

·        In what way do you calculate parameters (e.g. errors)???

·        line 182. p=3,5·10-5 ??? - I suggest write p=.0001 (significant statistically)

·        Figure 3,4,5. It would be great if authors present p-values on this figure between boxes

·        Lack of good state of the art. For example, authors should describe, that in gait analysis can be used Inertial Measurement Units. Of course, by obtaining kinematic parameters of each sensor you have error 2-3%. Look at the article DOI: 10.37190/ABB-01991-2021-05. Authors used IMUs and obtain joint angles. To improve article quality, I suggest write a few sentences about this methodology as an alternative.

·        Discussion: please compare results to the literature

·        Discussion: lack of limitation of the study and future study

Reviewer 2 Report

Dear Authors,

the paper is interesting and can give the new possibility for physiotherapist examination, however it needs some corrections.

Abstract:

Twenty-four stroke survivors with locomotion/gait disorder 18 (15 men and 9 women), please add the verb to the sentence.

81 please add the number of Ethical committee approval.

Methodology must be improved.

Results – how many steps did you count? Did you measure the same number of steps or did you ask for walking the specific distance.

182 and 183 instead of p=3∙10-12, please insert p=0,000

240-243 it is the explanation, which may be using in the Methodology or in the introduction.

Discussion needs to be rearrange – you should compare your findings with the other authors. Even if the papers are not strict the same, please show the other results which are published.

Best regards

Round 2

Reviewer 1 Report

Dear authors,

thank you very much for resending the article. My suggestions were included in the article.

Author Response

Thank you and we appreciate your time to review.
